# Potential of Whole-Body Vibration in Parkinson’s Disease: A Systematic Review and Meta-Analysis of Human and Animal Studies

**DOI:** 10.3390/biology11081238

**Published:** 2022-08-19

**Authors:** Y. Laurisa Arenales Arauz, Gargi Ahuja, Ype P. T. Kamsma, Arjan Kortholt, Eddy A. van der Zee, Marieke J. G. van Heuvelen

**Affiliations:** 1Department of Human Movement Sciences, University Medical Center Groningen, University of Groningen, 9713 AV Groningen, The Netherlands; 2Molecular Neurobiology, Groningen Institute for Evolutionary Life Sciences (GELIFES), University of Groningen, 9747 AG Groningen, The Netherlands; 3Department of Cell Biochemistry, University of Groningen, 9700 CC Groningen, The Netherlands; 4YETEM-Innovative Technologies Application and Research Centre, Suleyman Demirel University, Isparta 32260, Turkey

**Keywords:** exercise therapy, neuropathology, animal experimentation, human experimentation, motor skills, postural balance, gait, neuroinflammation, neurotransmitters, neurogenesis

## Abstract

**Simple Summary:**

Exercise has shown to have a positive impact on both motor and non-motor functions in Parkinson’s Disease patients. However, particularly in later stages of the disease, reduced cognitive function and motor capacity may lead to an inability to stay physically active. Therefore, alternative strategies for patients with Parkinson’s Disease are necessary to minimize burden for patients, their families and public health care. Whole-Body Vibration could be such an alternative. Whole-Body Vibration is an exercise or treatment method in which subjects are exposed to a mechanical vibration while sitting, standing or exercising on a vibrating platform. Whole-Body Vibration is currently used for physiotherapy, sports and rehabilitation purposes. Whole-Body Vibration treatment is interesting because it affects both the body and brain. The potential of Whole-Body Vibration for, specifically, Parkinson’s Disease patients should be clarified for further application. For this purpose, we conducted an extensive systematic review of the articles investigating the effects of Whole-Body Vibrations (1) on animals and humans with Parkinson’s Disease and (2) on neuropathological Parkinson’s Disease mechanisms. The results show some potential of Whole-Body Vibration for Parkinson’s Disease patients. The recommendations provided by this review can be used by researchers and rehabilitative practitioners implementing Whole-Body Vibration as a treatment for Parkinson’s Disease patients.

**Abstract:**

(1) Background: When the severity of Parkinson’s Disease (PD) increases, patients often have difficulties in performing exercises. Whole-Body Vibration (WBV) may be a suitable alternative. This systematic review aims to clarify if WBV shows potential as rehabilitative therapy for PD patients. (2) Methods: We searched several databases for controlled trials investigating the effects of WBV (1) on PD populations and (2) PD neuropathological mechanisms. We included both human and animal studies and performed meta-analyses. (3) Results: The studies on PD populations (14 studies) show an overall significant, but small, effect in favor of WBV (Hedges’ g = 0.28), for which the effects on stability (Hedges’ g = 0.39) and balance (Hedges’ g = 0.30) are the most prominent. The studies on the neuropathological mechanisms (18 studies) show WBV effects on neuroinflammation (Hedges’ g = –1.12) and several specific WBV effects on neurotransmitter systems, growth factors, neurogenesis, synaptic plasticity and oxidative stress. (4) Conclusions: The effects of WBV on human PD patients remains inconclusive. Nevertheless, WBV protocols with sufficient duration (≥3 weeks), session frequency (≥3 sessions/week) and vibration frequency (≥20 Hz) show potential as a treatment method, especially for motor function. The potential of WBV for PD patients is confirmed by the effects on the neuropathological mechanisms in mostly non-PD populations. We recommend high-quality future studies on both PD patients and PD mouse models to optimize WBV protocols and to examine the neuropathological mechanisms in PD populations.

## 1. Introduction

Parkinson’s Disease (PD) is a common neurodegenerative disease affecting millions of people worldwide, especially the aging population [1]. The hallmark pathogenesis of PD is defined by the progressive degeneration of dopaminergic nigrostriatal systems, manifesting in a wide range of clinical phenotypes, including motor (bradykinesia, tremor, muscle rigidity and postural instability) and non-motor (depression, anxiety, fatigue, cognitive dysfunction and dementia) functions [2,3]. The demographical transition from predominantly young to older populations, causes an increase in the prevalence and incidence of PD [4]. Therefore, the incorporation of rehabilitative strategies for PD is necessary to minimize the burden for the patient, their families and public health care [1].

The treatment guidelines for PD management include physical exercise as an adjuvant to pharmacological treatments and surgery [5]. Exercise has shown to have a positive impact on both motor and non-motor functions by ameliorating clinical symptoms and modulating brain maintenance and plasticity in PD patients [6,7]. However, particularly in the later stages, reduced cognitive function and motor capacity may lead to the inability to stay physically active [8]. When PD patients are not (cap)able of performing exercise, an alternative approach is needed to fully benefit from its positive effects. Whole-Body Vibration (WBV) may be a suitable alternative.

WBV has been considered as an alternative for active exercise. A major advantage of WBV is that it can be used by people who cannot or have severe difficulties to participate in active exercise programs. Participants may be too frail or may not have the physical condition to perform active exercise, next to motivational or mental problems. In general, WBV does not take much effort; WBV procedures are relatively brief, and can be performed indoors at any moment of the day with limited or without supervision. The benefits of WBV on motor performance [9,10,11,12] are, to a large extent, comparable to those reported after active exercise; hence, it can be viewed as a form of passive exercise.

During WBV, a vibration is transmitted through the body by maintaining an active (e.g., performing squats) or passive posture (e.g., sitting or lying) on a mechanically vibrating platform ([13] and references therein). The vibrations can be categorized into harmonic or random oscillations with synchronous vertical motions or motions around the sagittal axis [14] (Figure 1). The vibrations are defined further by their frequency (generally between 5–60 Hz), amplitude or peak-to-peak displacement (generally between 0.5–4 mm peak-to-peak) and temporal aspects, such as number (generally between 1–10 bouts) and durations of bouts (generally between 30 s and 10 min, also depending on the number of bouts).

Patients with PD may be potential beneficiaries of WBV in view of the multidimensional effects of WBV (see, for review, [15]). Several studies have shown that WBV improves muscle strength [16], postural control [17], bone health [18] and balance [19] in older adults. It has also been suggested that WBV can improve bone fractures [20] and wound healing [21] in animals. However, not much is known about the impact of WBV on foot health. In PD patients, proper foot health is in general relevant for good motor performance [22,23]. In people with type II diabetes mellitus, a protocol of eight weeks of WBV improved foot health [24], but so did the controls that followed the same procedure without the actual vibrations. Hence, the improvement did not depend on the actual vibrations, and it should be stressed that WBV did not interfere with foot health improvements. Furthermore, both animal and human studies [25,26,27] provide empirical support for a potential value of WBV in improving cognition. Moreover, WBV research has been performed in health-compromised individuals with both motor and non-motor deficits, such as stroke [28], multiple sclerosis [29,30] and dementia [31], indicating that WBV intervention is feasible in health-compromised patients.

In line with this, reviews have been published on the effects of WBV in PD patients [32,33,34,35,36,37]. However, these studies have a limited scope of view because they are solely based on human research with outcome measures restricted to the sensorimotor and functional levels. Given the wide range of clinical phenotypes ranging from mild motor disturbances to severe motor, cognitive, affective and sleep disturbances [2], an overarching perspective of the potential of WBV exercise in PD patients is needed.

The purpose of this study is to provide a systematic overview with meta-analysis in order to clarify if WBV shows potential as rehabilitative therapy for PD patients. We examine two research questions: (1) What are the effects of WBV on humans and animal models with PD? (2) Are there potential ameliorative effects of WBV on the neuropathological mechanisms of PD? Outcome measures on cellular, brain and functional levels are included to provide a complete perspective.

## 2. Methods

We followed the PRISMA guidelines for reporting the systematic reviews [38]. This review was registered in Prospero under ID number CRD42022351823.

### 2.1. Information Sources

We conducted a systematic search using the following digital online databases: PubMed, EMBASE, Scopus and Web of Science. These databases were accessed online in March 2021.

### 2.2. Search Strategy

The search strategy was designed to examine (1) the effects of WBV on humans and animal models with PD, and (2) the potential ameliorating effects of WBV on neuropathological mechanisms of PD. Key search terms related to WBV training were combined with terms specified to PD itself (research question 1) or terms related to the neuropathology of PD (research question 2).

The search terms for the neuropathology of PD were determined by empirical evidence of dopaminergic and non-dopaminergic neurotransmitter losses [2,39,40,41], programmed cell death [42], oxidative stress [2,43] and Lewy Bodies [44] in humans and PD animal models. Furthermore, ‘cellular’, ‘brain’ and “blood” search terms were added to amplify the search for potential mechanisms.

We used the following key search terms for PubMed and adapted this strategy for the other databases (Table A1): (“Whole Body Vibrat*”[tiab] OR “Vibration Therap*”[tiab] OR “Vibration exercise*”[tiab] OR “Vibration Training*”[tiab]) AND ((“Parkinsonian disorders”[Mesh] OR “Parkinson*” [tiab] OR “MPTP” [tiab]) OR (“Parkinson Disease/physiopathology”[Mesh] OR “Neurotransmitter Agents”[Mesh] OR “Neurotransmitter Agents” [Pharmacological Action] OR “Serotonin”[Mesh] OR “Dopamine”[Mesh] OR “Cytokines”[Mesh] OR “Nerve Growth Factors”[Mesh] OR “Calcium”[Mesh] OR “gamma-Aminobutyric Acid”[Mesh] OR “Blood Circulation”[Mesh] OR “Blood”[tiab] OR “Brain*” [tiab] OR “Neural Activation*”[tiab] OR “neurotransmitter*” [tiab] OR “serotonin*” [tiab] OR “dopamine*”[tiab] OR “acetylcholine*”[tiab] OR lewy bod*[tiab] OR “Oxidative stress*”[tiab] OR “Cytokine*”[tiab] OR “neurotrophic factor*” [tiab] OR “calcium*”[tiab] OR “gamma-amminobutyric acid*”[tiab] OR “bdnd” [tiab] OR “gaba”[tiab] OR “gdnf” [tiab])).

### 2.3. Study Selection

We removed duplicates from the records identified by the databases with the Mendeley reference management tool. Two authors (Y.L.A.A. and G.A.) screened the titles and abstracts based on the eligibility criteria. Subsequently, one author (Y.L.A.A.) screened full texts for inclusion in the qualitative synthesis based on the same eligibility criteria. In case of doubts, the other author (G.A.) screened the full texts as well. Disagreements on the overall article selection were resolved through mutual consensus by all co-authors. Finally, we manually searched the reference list of each selected article and review articles to identify other relevant articles.

### 2.4. Eligibility Criteria

We followed the PICO framework [45] to formulate the eligibility criteria. The intervention (I) and control (C) eligibilities were the same for both research questions. The criteria of population (P) and outcome (O) were specified separately for each research question. Starting with the intervention (I), articles were included if (1) WBV was applied as a mechanical vibration transmitted through a platform or chair, (2) the design consisted of randomized or non-randomized controlled trials with single and/or multiple WBV treatment sessions and (3) acute, short- and/or long-term effects were measured. Related to the control (C), the comparison condition or group consisted of no treatment, pseudo WBV, sham WBV (exposure of WBV to a negligible low frequency, or noise only) or another form of exercise. Furthermore, we included articles with a publication date till March 2021. Articles were excluded if (1) any type of vibration therapy other than WBV (e.g., vibration pads, ultrasound, electrical stimulation) was examined and, (2) articles were not written in English or (3) articles were review articles.

Separate criteria were designed for each research question regarding the population (P) and outcome (O) measures. To examine the effects of WBV on humans and animal models with PD (research question 1), each selected article had to satisfy the following criteria: (1) the study population consisted of patients diagnosed with PD or PD animal models, (2) no limitations were set on disease severity, age or medication and (3) no restrictions were set on outcome measures.

To examine the potential ameliorating effects of WBV on neuropathological mechanisms (research question 2), articles were included if (1) humans or animal models were examined, and (2) outcome measures were related to neuropathological mechanisms of PD as described in the key search terms.

### 2.5. Data Extraction

One author (Y.L.A.A.) extracted data from each selected article. For each research question, the characteristics and results of the study were presented separately. Characteristics of the studies included the target population, number of participants, intervention and control groups, study design, intervention duration and WBV specifications. In accordance with the guidelines of van Heuvelen et al. (2021), the WBV specifications included type of vibration (synchronous vertical or side-alternating), frequency (Hz), amplitude (mm), temporal aspects (number of sessions per week, number of bouts per session, bout duration) and posture(s).

The results were categorized into the specific outcome measures and into animal and human results. The results included the examined areas with specific outcome measures, pre-test and/or post-test results, within-group effects, between-group effects and main findings. If the authors only presented graphed data, we extracted the data points with a validated web-based tool (WebPlotDigitizer) [46]. To create a general consensus regarding the terminology, we referred to “acute effects” when outcome measures were measured following one WBV session, whereas “chronic effects” referred to outcomes measured over a longer period of time following WBV intervention.

### 2.6. Quality Assessment

One author (Y.L.A.A.) assessed the methodological quality of the studies included. The standardized and validated PEDro scale was used to evaluate the external and internal validities of the controlled, clinical, human studies [47]. Trials were rated with a score ranging from 0–11 with higher scores representing a superior methodological quality. A Scores ranging from 0 to 4 indicated “poor”, 5–6 “fair”, 7–9 “good” and 10–11 “excellent” qualities.

The ARRIVE essential 10 guidelines were used to determine the methodological quality of animal studies [48]. The aforementioned guidelines ensured the rigorous design and reporting of animal research with a score ranging from 0–18. Scores ranging from 0–7 indicated “poor”, 8–12 “fair”, 13–15 “good” and 16–18 “excellent” qualities.

### 2.7. Data Synthesis for Meta-Analyses

We used Review Manager Software (Revman) 5.4.1 to conduct a meta-analysis [49] for each research question separately. We used the inverse variance method with a random-effect model. Hedges’ adjusted g effect sizes with 95% confidence intervals (CIs) were calculated for each continuous variable (benchmarks 0.20, 0.50 and 0.80 for, respectively, small, medium and large effects). In case of a two-group pre-post design study, we included the post-pre or pre-post (for inversed scales) difference scores with the pooled standard deviations (square root of the mean of pre- and post-variances) into the analyses. We performed subgroup analyses for each outcome domain. If a study included more than one outcome measure, we incorporated each outcome measure separately into the analyses. I^2^ was interpreted as the main heterogeneity measure with the following benchmarks: 0–40, not important; 30–60, moderate; 50–90, substantial and 75–100, considerable heterogeneity [50]. 

## 3. Results

### 3.1. Study Selection

The flowchart of the article selection process is presented in Figure 2. The initial database searches produced a total of 1904 citations. Two additional articles were identified through other sources. Among 1904 articles, 106 articles were identified as potentially relevant and were subjected to full-text screening. A total of 32 articles was sufficient for meeting the eligibility criteria for research questions 1 (14 studies) and 2 (18 studies). The data for 29 studies were synthesized in the meta-analyses. The interrater reliability revealed a 91% agreement between the two authors (Y.L.A.A. and G.A.).

### 3.2. Effects of WBV on PD Populations

The effects of WBV on PD populations were investigated in 14 studies. Thirteen studies included human participants diagnosed with PD [51,52,53,54,55,56,57,58,59,60,61,62,63] and one study in a PD mouse model following brain decapitation [64].

Table 1 presents the characteristics of the studies. Twelve studies had an RCT design and two studies used a non-RCT design, including a non-randomized clinical trial and a cross-over study. In total, 476 human participants diagnosed with PD (mean age: 66.1 years; M/F: 299/149, no gender was reported for 28 participants) and 26 1-Methyl-4-phenyl-1,2,3,6-tetrahydropyridine (MPTP) mice, a commonly used animal model for PD, were included. The average severity of the disease in human participants ranged from mild to moderate bilateral dysfunctions without serious balance disability (range mean Hoehn and Yahr stage: 2.0–3.3) [65]. The control intervention, intervention duration and WBV parameters varied across all studies. The control interventions comprised sham vibration [53,54], placebo [51,59,61,63], rest [56,57], listening to music [55], moderate walking [52] and conventional (physical) therapies [58,60,62]. The intervention duration varied from one day to four weeks. Nine studies used vertical and four studies side-alternating (one study not reported) vibrations. The number of WBV sessions ranged from 1 to 28. WBV frequency ranged from 6 Hz to 30 Hz and amplitude: 2–14 mm. The number of bouts per session varied between 1 to 15 per session, and the bout duration ranged from 20 s to 15 min.

The methodological quality of the human studies was considered “fair” [52,56,57,62], “good” [51,54,55,58,59,60,63] or “excellent” [53,61] according to the PEDro scale (Table A2). The animal study [64] was considered of “good” quality according to the ARRIVE guidelines (Table A3). Appendix A presents an overview of the psychometric properties of the measurement instruments used in the human PD studies.

Table 2 presents an overview of the main findings. In humans, ten studies found better scores post WBV vs. pre WBV for several outcome measures, including overall motor function [53,54,55,56,60], mobility [59], fall risk [53], gait [53,59,60], balance [51,58,59,60], postural stability [52,53,54], flexibility [61], non-motor symptoms [55], anxiety [55], depressive symptoms [55], quality of life [55], fatigue [55] and metabolic effects [62]. Four studies revealed a superior beneficial effect of WBV compared to the control group or control condition on specific outcome measures. These outcome measures were related to postural stability [52,59], balance [58] mobility [59], flexibility [61] and rigidity [59]. In MPTP mice, BDNF and dopamine levels in the striatum significantly increased following chronic WBV, compared to the control intervention [64]. Hedges’ g effect sizes were generally small to moderate, with seven outcome measures revealing large effects. Figure 3 shows the results of the meta-analysis. Overall, WBV appeared to be beneficial for PD populations with a small effect size (Hedges’ g = 0.28). On the level of outcome domains, the accumulated effects were significant for balance (Hedges’ g = 0.30), stability (Hedges’ g = 0.39) and non-motor functions (Hedges’ g = 0.62). Heterogeneity was low, except for non-motor functions (I^2^ = 66%).

### 3.3. Effects of WBV on PD Neuropathological Mechanisms

Potentially ameliorating effects of WBV on PD neuropathological mechanisms were investigated in eighteen studies: seven studies on non-PD human populations [66,67,68,69,70,71,72] and eleven on non-PD animal populations [25,73,74,75,76,77,78,79,80,81,82]. Unlike Zhao et al. (2014; see Table 1 and Table 2), we did not obtain studies on the PD-ameliorating mechanisms of WBV in PD populations. Table 3 presents the characteristics of the included studies. Two cross-over studies and 16 RCTs were performed. A total of 147 human participants (range of average age: 23,4–73 years, M/F: 34/102; no gender was reported for 9 participants) and 321 healthy or diseased modeled mice or rats were included. Intervention duration varied between 1 day and 12 weeks, 12 studies applied vertical vibrations and 3 studies side-alternating vibrations. The vibration frequency varied from 10 to 45 Hz. The amplitude ranged between 2.5 and 4 mm in human studies and between 0.014 and 5 mm in animal studies. The number of sessions varied from 1 single session to 7 sessions/week during 12 weeks, the number of bouts per session ranged from 1 to 10 bouts and the bout duration from to 20 s to 240 min.

The methodological quality of the human studies was “fair” [67,68,69,70] or “excellent” [66] according to the PEDro scale (Table A2). All animal studies met at least half of the criteria (18 items) of the ARRIVE guidelines with methodological quality scores ranging from 9 to 15 (Table A3), which were considered as “fair” [73,74,77,81,82] or “good” [25,75,76,78,79,80].

Table 4 presents the effects of WBV on neuropathological mechanisms. Neurotransmitter release following chronic WBV was reflected in increased plasma noradrenalin in humans [72]. In specific brain regions in the animals, acute WBV increased cerebral dopamine [77], noradrenaline [81] and serotonin [81]. Chronic WBV increased cholinergic activity further [80].

Several studies showed inflammatory markers to be responsive to WBV. Chronic WBV effects were reflected in the lower levels of pro-inflammatory markers (TLR2, TLR4, TNFa [68]; caspase 1, IL-1β [79]), but acute WBV effects were expressed in higher levels (TNFa [70]; sTNFR1, for healthy women only [69]). Three studies included anti-inflammatory markers. Lower levels were observed following acute WBV (adiponectin and sTNFR2 in women with fibromyalgia only [69]) and chronic WBV (ASC [79]). Only one study of acute WBV revealed higher levels of inflammatory markers (IL-10 [70]).

Regarding the growth factors, a significant upregulation of BDNF following chronic WBV was observed in humans [66] and animals (BDNF as well as pTrK-B [79]; BDNF [78]) Additionally, higher levels of VEGF were observed following chronic WBV in humans [70]. Chronic WBV increased IGF-1 levels further in CRS rats [78], but decreased in mice models with atherosclerosis [82].

The research on brain-related changes during WBV showed an increased cortical activation in humans [71]. In animals, chronic WBV reduced neurodegeneration [78], increased neurogenesis [75] and increased synaptic plasticity in the hippocampus [74]. No changes in brain glucose uptake following chronic WBV were observed [25]. For oxidative stress markers, acute WBV improved the oxidant and antioxidant parameters in humans [67], and chronic WBV attenuated oxidative stress in mice [76].

Figure 4 presents the results of the meta-analysis. Overall, and in each outcome domain, except for brain-related changes, heterogeneity appeared considerable (81 ≤ I^2^ ≤ 91%). For the brain-related changes, heterogeneity was substantial (I^2^ = 71%). The accumulated effect size was only significant for inflammatory markers with a considerable reduction in levels following WBV (Hedges’ g = −1.12).

## 4. Discussion

This systematic review aimed to determine if WBV training exhibited potential as treatment for PD patients. In total, 32 human and animal studies were included. Fourteen studies investigated the effects of WBV on PD populations and 18 studies investigated the potential ameliorating effects of WBV on neuropathological mechanisms of PD in non-PD populations. Prior systematic reviews only focused on human PD studies with outcomes restricted to the domains of motor impairments, balance, gait and/or mobility [32,33,34,35,36,37]. New insights into the etiology of PD and mode of actions of WBV required the reconsideration of the potential of WBV as a valuable intervention for PD. With the inclusion of animal studies, non-motor outcomes in PD and PD-relevant neuropathological mechanisms in non-PD populations, we extended the search considerably. Moreover, our broader view will facilitate the identification of the relevant, but unexplored, outcome domains.

### 4.1. WBV in Human PD Populations

At present, WBV studies on human PD populations (13 studies) do not provide a conclusion as to whether WBV is effective. Although the vast majority of the studies revealed pre- versus post-improvements following WBV, only four studies provided statistical evidence for the superior beneficial effects of WBV vs. control [52,56,58,59], while four studies did not report on WBV vs. control [51,54,61,62]. The meta-analysis revealed a significant, but small, effect in favor of WBV (Hedges’ g = 0.28, *p* < 0.0001). However, given the considerable variation of control interventions, WBV protocols and settings, and outcome measures, WBV’s lack of a convincing superiority over control should be regarded with caution.

The control interventions include active (e.g., walking or physical therapy) and non-active (sham or placebo) interventions. Active-control interventions may lack contrast with WBV training, limiting the chance of finding differential effects of WBV vs. control. However, a comparison to the conventional treatments may support the applicability of WBV. The fact that WBV and physical therapy have comparable beneficial effects on balance, gait and motor PD symptoms [60] can be considered as promising, since WBV is easier to apply and requires less effort [62]. In line with this, WBV is well tolerated [83] and feasible [84] to apply to patients with disorders.

The majority of the studies included motor outcome measures, including balance, stability, mobility, gait (with a variety of tests) or specific motor PD symptoms. The subgroup analysis of the meta-analysis showed significant overall effects for balance (Hedges’ g = 0.30, *p* < 0.05) and stability (Hedges’ g = 0.39, *p <* 0.01), but not for mobility (Hedges’ g = 0.32, *p >* 0.05, gait (Hedges’ g = 0.19, *p >* 0.05) or specific motor PD symptoms (Hedges’ g = 0.18, *p >* 0.05). These values obtained for Hedges’ g are comparable to those observed in another recent review of the meta-analysis [32]. On the level of individual studies, the significant beneficial effects of WBV vs. control were observed for balance/stability [52,58,59], mobility [59] and motor PD symptoms [56]. In addition, Kaut et al. (2011) observed improvements in motor PD symptoms (including rigidity, postural stability and bradykinesia) following WBV and not after sham vibration with a considerable effect (Hedges’ g = 0.76), but without reporting statistical testing. Only one study included non-motor outcomes related to depression, anxiety and fatigue, and observed similar pre-post improvements for WBV and listening to music (Hedges’ g =, respectively, –0.21, –0.10 and –0.07) [55]. Based on the current studies, we cannot conclude that heterogeneity in outcome measures plays a major role, but non-motor outcomes, including cognitive function, are hardly represented.

WBV protocols and settings varied considerably across the studies. In four studies, the short-term effects were examined following a single session of WBV with mixed results. The other studies focused on the chronic effects. Overall, longer intervention durations (3–5 weeks) with sufficient session frequencies (3–4 sessions per week) might be more promising, although one study with a duration of 5 weeks and 2–3 sessions per week did not observe any effects of WBV compared to the placebo [63]. The latter result might be related to the relatively low vibration frequency (6Hz), although a study with comparable duration, session frequency and vibration frequency [59] revealed significant positive effects of WBV vs. placebo. Only one study applied several vibration frequencies and concluded that higher frequencies seem to present a greater improvement [61]. Indeed, Guadarrama-Molina and co-authors (2020) applied a frequency of 20 Hz during 5 weeks and observed positive effects of WBV vs. conventional therapy, and Ebersbach et al. (2008) applied a frequency of 25 Hz during 3 weeks and observed similar effects of WBV and physical therapy. The stronger effects of higher vibration frequencies are in agreement with results findings of a review of COPD patients [85] and may be explained by enhanced neuromuscular activity [86].

### 4.2. WBV Related to the Neuropathological Mechanisms of PD

None of the studies on human PD patients investigated the neuropathological PD mechanisms, and solely one animal study on PD mice investigated the neuropathological mechanisms. However, the neuropathological mechanisms of PD in non-PD animal (11 studies) or human (7 studies) populations may provide further insight into the potential of WBV for this patient group. Animal studies are valuable [87] because biological processes can be examined in detail, whereas examining mechanisms in humans is expensive, stressful for the patient or even impossible/unethical. In addition, in animals, processes can be studied in less time because of the shorter lifespan of frequently used species (e.g., mice and rats), and within strains, genetic variations and variations in lifestyle regimes are considerably reduced.

Only one study included a PD population, in which MPTP mice were exposed to WBV during four weeks on a daily basis [64]. The vibration training almost completely prevented the MPTP-induced loss of dopamine neurons in the substantia nigra and upregulated BDNF. This is an important result, since it suggests that WBV could act upon the primary cause of PD. It may also explain the improvement to specific motor symptoms (UPDRS-III score) following a WBV intervention [53,54,55,60,63].

All other WBV studies related to neuropathological PD mechanisms were conducted on non-PD populations. Overall, the meta-analysis presented no effect of WBV (Hedges’ g = 0.04, *p >* 0.05), but this observation should be interpreted in light of considerable heterogeneity (I^2^ = 86%). Therefore, it is more meaningful to consider the subgroup results or even the individual study results.

For the neurotransmitter domain, the meta-analysis resulted in an overall moderate to large effect with higher neurotransmitter levels following WBV, but the effect was non-significant (Hedges’ g = 0.62, p = 0.09). However, considerable heterogeneity also played a major role (I^2^ = 81%) here, which caused us to consider specific neurotransmitters and individual studies. Two non-PD studies investigated the effects on dopaminergic systems. In these studies, Wistar rats were exposed to single bouts of WBV. Following 90 min of WBV, the dopamine levels increased in the nucleus accumbens (part of ventral striatum) and frontal cortex (20 Hz; [77]), but dopamine levels in the cortex and striatum appeared unaffected after 240 min (20 Hz; [81]. Although the first study may support a beneficial effect on the dopaminergic system, the value for exercise studies in which the WBV bouts are much shorter is limited. These long bout-duration studies mainly focus on safety issues regarding “bad vibrations” associated with detrimental effects [15,88].

Other neurotransmitter systems are affected by WBV as well, with enhanced levels of epinephrine and norepinephrine in human blood plasma levels following a single session of WBV (26 Hz and 10 × 60 s; [72]). Following lengthy exposure to WBV (240 min), norepinephrine levels decreased in the specific parts of the brain of Wistar rats [73,81], while serotonin levels also increased [81]. Heesterbeek et al. (2017) suggested that a five-week WBV intervention (30 Hz) activated the cholinergic system in C57BI/6J mice. Altogether, WBV has the potential to affect neurotransmitter systems. This may be promising for PD patients, since not only is the dopaminergic system impaired in PD, but also the adrenergic, serotonergic and cholinergic systems [2]. However, this should be confirmed in the studies including PD patients.

Chronic neuroinflammation is one of the hallmarks of PD pathophysiology and therefore an important therapeutic target [89]. Overall, WBV resulted in reduced levels of inflammatory markers (Hedges’ g = –1.12, *p <* 0.01), but with considerable heterogeneity (I^2^ = 91%). Three human and two animal studies examined the effects of WBV on pro-inflammatory markers (e.g., cytokines, such as TNFa and interleukin 1β, Toll-like receptors TLR2 and TLR4, tumor necrosis factor TNFa or receptor sTNFR1), anti-inflammatory markers (e.g., adiponectin, sTNFR2, interleukin 10) or markers with both pro- and anti-inflammatory properties (interleukin 6). The studies on the chronic effects of a WBV intervention consistently revealed significant decreases in several pro- [68,79] and anti-inflammatory markers [79,82]. Acute effects, however, showed enhanced levels of some pro-inflammatory markers [69,70] with a decrease [69] or increase [70] in anti-inflammatory markers. These results suggest that WBV may evoke an acute inflammatory response. An acute inflammatory response is an essential and protective response in, for example, injured tissues. This response can restore the tissues to their pre-injury state [90]. The chronic inflammatory response may indicate that the inflammation balance is altered, with an overall lower level of chronic inflammation.

Indeed, a recent paper confirmed that WBV attenuated traumatic brain injury-related damage through the regulation of neuroinflammation [91]. Whether this is also true for PD should be examined in either research on animals or humans.

Changes in the growth factors (BDNF, IGF-1) are considered to be an underlying mechanism of the cognitive effect of exercise [92]. The meta-analysis showed an overall moderate to large, but non-significant, increase in growth factor level after WBV (Hedges’ g = 0.67, *p >* 0.05) with considerable heterogeneity (I^2^ = 86%). Chronic increases in plasma BDNF [66] and brain BDNF [79] concentrations were observed after a WBV intervention [66,79], but not for a single session [69]. For IGF-1, the results are mixed with chronically increased [78] or decreased [82] levels. The difference may be explained by heterogeneity in frequency (respectively, 30 Hz and 15 Hz) or animals (respectively, mice and rats) used. The potential of WBV to affect the brain is supported further by some animal studies, showing increased hippocampal synaptic plasticity [74], increased neurogenesis [75] and decreased neural degeneration [78]. The meta-analyses revealed a small to moderate, but non-significant, effect (Hedges’ g = 0.35, *p >* 0.05) with substantial heterogeneity (I^2^ = 71%). Nevertheless, since PD goes together with cognitive decline [2,43] these findings may be promising, but should be confirmed.

Oxidative stress, an imbalance between increased levels of reactive oxygen species (free radicals) and low activity of antioxidant defense, is considered to modulate the development of PD [93]. Only two studies examined WBV in relation to oxidative stress markers resulting in an overall less than small effect (Hedges’ g = 0.11, *p >* 0.05) with considerable heterogeneity (I^2^ = 83%). Notwithstanding, the findings from both a single session of WBV in humans (women with fibromyalgia; [67]) and a 12-week WBV intervention in mice [76] support a beneficial effect of WBV on oxidative stress. However, the translational value to PD is unknown, so these findings should be interpreted with caution.

### 4.3. General Considerations

In summary, based on human studies on PD patients, we cannot definitively confirm the effectiveness of WBV for PD patients considering the significant, but low, overall effect size. However, there may be potential for WBV protocols with sufficient duration (at least three weeks), session frequency (at least three sessions per week) and vibration frequency (at least 20 Hz), especially for outcomes related to motor function. Studies on neuropathological mechanisms of PD further confirm the potential of WBV to reduce neuroinflammation and promote specific neurotransmitter systems, increase the expression of specific growth factors, increase neurogenesis and synaptic plasticity and reduce specific markers of oxidative stress.

However, all results should be interpreted with caution. Optimal settings are not definitively defined yet, since high-quality studies with a direct comparison of different settings are still scarce. Furthermore, optimal settings may vary across desired outcomes and individual patients. Whether the effects depend on posture remains uncertain, since the majority of the human studies used a static semi-squat posture. Furthermore, the peak-to-peak displacement (or amplitude) does not vary strongly across the studies. However, in most studies, the peak-to-peak displacement of the vibrations remains unclear, since they are not defined and/or not verified with actual measurements. Since the manufacturer’s settings and measurement of the vibration often lead to different outcomes, it is important to report adequate information here [13]. With current studies, we cannot yet elaborate on the moderating effect of peak-to-peak displacement. Whether the effects depend on the type of vibrations (vertical vs. side-alternating) is still unknown since there is a paucity of studies with side-alternating WBV in PD populations. Although several outcome measures validated for PD were used (i.e., UPDRS-III, 8 m or 10 m walk test, Tinetti scale and Berg balance scale), other outcome measures were only validated in other populations (e.g., timed up and go test, BDI depression scale, ISQ anxiety scale or FFS fatigue scale) or, as far as we know, not yet validated (e.g., freezing test and step–walk-–turn test. Subsequently, the studies regarding the neuropathological mechanisms of PD in PD populations is very limited, with only one study on MPTP mice [64]. The translational value of study results obtained from non-PD to PD populations and from animal to human populations in the context of WBV is still unsure. In addition, several studies with long WBV exposure, investigating “bad vibrations”, are less relevant for WBV in the context of exercise or treatment. Finally, several PD relevant outcomes have not yet been investigated, such as cognitive function in PD patients, Lewy bodies or alpha-synuclein aggregation.

Compared to the previous systematic reviews [32,33,34,35,36,37], our broader and novel scope resulted in a more detailed identification of WBV settings and protocols delivering further perspectives for PD patients. Furthermore, it strengthened the evidence of the potential of WBV considering the effects observed on neuropathological mechanisms, such as neuroinflammation and specific neurotransmitter systems. Finally, it resulted in the identification of (relatively) unexplored research domains (e.g., neuropathological mechanisms in PD and non-motor outcomes, such as cognition).

### 4.4. Recommendations for Further Research

For further research, we recommend high-quality human studies on PD patients with an intervention duration of at least three weeks with at least three sessions per week, a vibration frequency of at least 20 Hz and with sufficient contrast between WBV and control group(s) (e.g., WBV, sham WBV and normal care). Furthermore, we recommend RCTs with different levels of frequency and/or peak-to-peak displacement and with side-alternating vibrations in order to make progress in identifying the optimal settings. We recommend the use of outcome measures validated for PD and adding outcomes related to non-motor functions, such as cognitive and affective functions, next to general motor function and PD-specific motor symptoms. We recommend animal studies using PD-mouse models to investigate the PD-relevant mechanisms in more detail and to investigate different durations and settings relevant for exercise and treatment with WBV, in order to shape human research to identify the optimal settings. Finally, we recommend using the guidelines of van Heuvelen et al. (2021) [13] to promote correct, complete and consistent WBV reporting.

### 4.5. Recommendations for Sports and Rehabilitation Practices

Based on the current knowledge, we cannot recommend WBV as a primary treatment for PD patients in the exercise domain. If moderate-to-high intensity conventional exercise, such as resistance and aerobic training, is possible, this kind of exercise still has priority. However, for those who are not able to perform active exercises, WBV may be a valuable alternative. Furthermore, WBV can be performed in addition to conventional exercises to enhance the beneficial effects of exercise for PD patients. Sufficient intervention duration, session frequency and vibration frequency are important as well as guidance by a qualified trainer for correct performance and to promote efficacy and adherence [94].

### 4.6. Limitations

This systematic review had some limitations. First, we limited our review to animal and human studies, excluding cell culture studies. The inclusion of cell culture studies may provide additional insights into the potential and underlying molecular mechanisms of WBV for PD patients. More specifically, the use of human-induced pluripotent stem cells (hiPSCs) isolated from PD patients might be of great benefit. Second, with regard to non-PD populations, we restricted our search to direct PD neuropathological mechanisms. We considered indirect mechanisms (e.g., mechanisms on muscle level indirectly affecting the brain) as beyond the scope of this review. Finally, although we performed a meta-analysis on the effects on neuropathological PD mechanisms in non-PD populations, the generally considerable heterogeneity required us to interpret the overall results with caution.

## 5. Conclusions

Our review revealed the significant but minor beneficial overall effect of WBV on motor and non-motor outcomes in PD patients. Therefore, the potential of WBV for PD patients seemed limited, although specific protocols performed with sufficient frequency and duration may show potential and legitimizes further research. The overall reduced levels of neuroinflammation and specific WBV effects on other PD-relevant neuropathological mechanisms observed in non-PD populations strengthen the urgency for further research into the effects of WBV on PD patients. The recommendations for practice and further research are summarized in Table 5.

## Figures and Tables

**Figure 1 biology-11-01238-f001:**
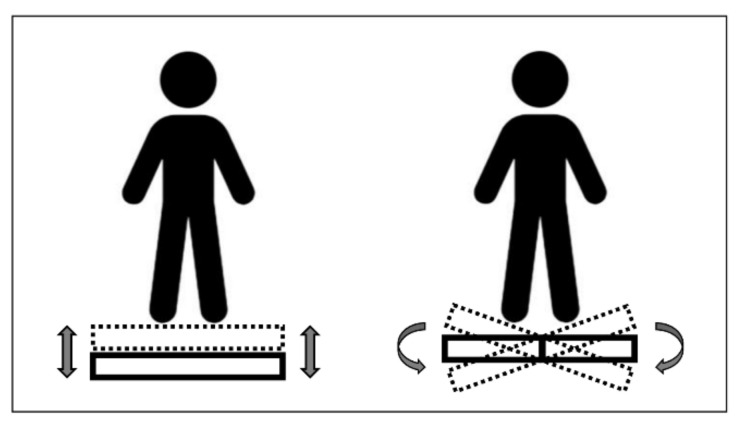
Vertical (**left**) versus side-alternating (**right**) vibrations in standing position.

**Figure 2 biology-11-01238-f002:**
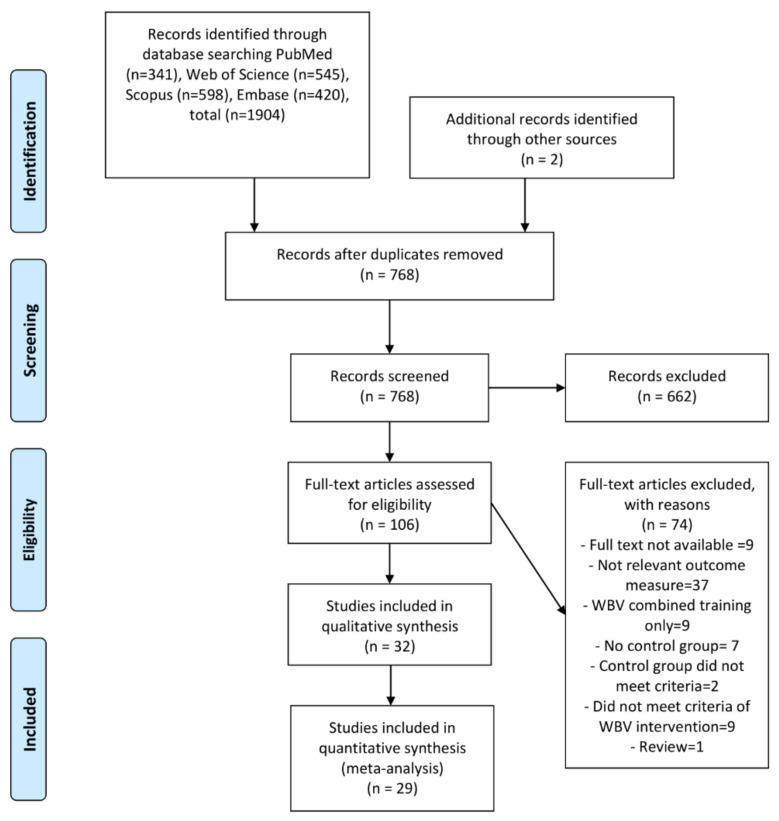
Flow diagram of article selection process.

**Figure 3 biology-11-01238-f003:**
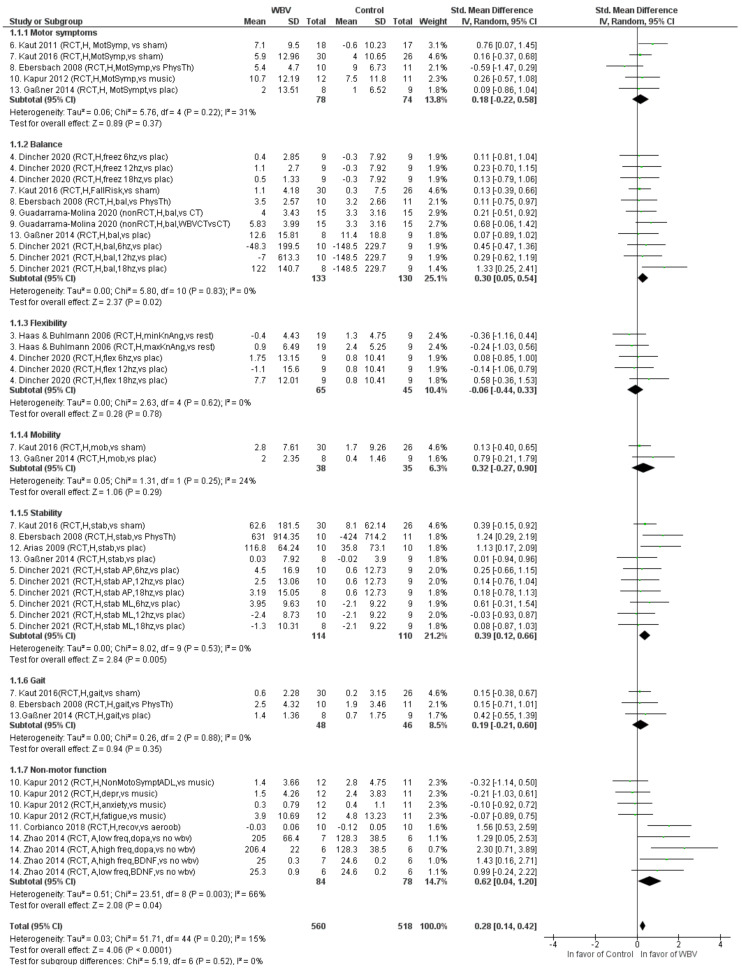
Forest plot with Hedges’ g effect sizes and 95% confidence intervals for motor and non-motor symptoms in PD populations (research question 1). Abbreviations: RCT = randomized controlled trial; H = human study; A = animal study; MotSymp = motor symptoms; Sham = sham vibration; PhysTh = physiotherapy; Plac = placebo; Freez = freezing; Bal = balance; CT = conventional therapy; MinKnAng = minimum knee angle; MaxKnANG = maximum knee angle; Flex = flexibility; Mob = mobility; Stab = stability; AP = anterior–posterior; ML = medio-lateral; NonMotoSymptADL = non-motor symptoms during activities of daily living; Aerob = aerobic treadmill training; Depr = depression; Recov = recovery phase; Dopa = dopamine; BDNF = brain-derived neurotrophic factor.

**Figure 4 biology-11-01238-f004:**
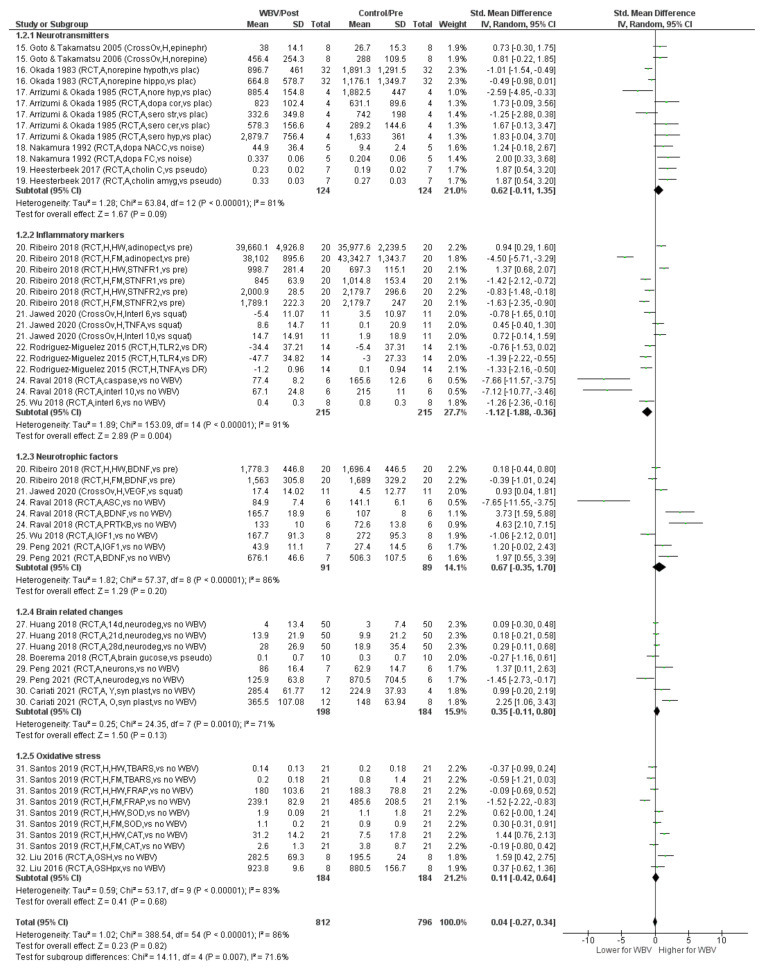
Forest plot with Hedges’ g effect sizes and 95% confidence intervals for the specified outcome measures regarding neuropathological PD mechanisms in non-PD populations (research question 2). Abbreviations: RCT = randomized controlled trial; CrossOv = cross-over design; H = human study; A = animal study; Epinephr = epinephrine; Norepine, Nore = norepinephrine; Hypoth, Hyp = hypothalamus; Hippo = hippocampus; Dopa = dopamine; Cor,C = cortex; Sero = serotonine; STR = striatum; Cer = cerebellum; NACC = nucleus accumbens; FC = frontal cortex; Cholin = cholinergic system activity; Amyg = amygdala; Pseudo = pseudo WBV; HW = healthy women; FM = women with fibromyalgia; Adinopect = adinopectine; STNFR1 = soluble tumor necrosis factor receptor-1; STNFR2 = soluble tumor necrosis factor receptor-2; Interl 6 = interleukin-6; TNFA = tumor necrosis factor alpha; Interl 10 = interleukin 10; TLR2 = toll-like receptor 2; DR = daily routine; TLR4 = toll-like receptor 4; BDNF = brain derived neurotrophic factor; VEGF = vascular endothelial growth factor; ASC = adipose derived stem cells; PRTKB = phospho-receptor tyrosine kinase; IGF1 = insulin-like growth factor-1; Neurodeg = neurodegeneration; Syn plast = synaptic plasticity; Y = young; O = old; TBARS = thiobarbituric acid reactive substances; FRAP = ferric reducing ability of plasma; SOD = superoxide dismutase antioxidant enzymes activity; CAT = catalase; GSH = glutathione.

**Table 1 biology-11-01238-t001:** General overview of study characteristics regarding the effects of WBV on PD patients or PD animal models (N = 14 studies).

Reference	Design	Sample	Disease Severity (Mean H&Y ^1^)	Groups (n)	Intervention Duration	Vibration Protocol			
						Type (Device, Vibration Type)	Temporal Aspects ^2^	Intensity (Frequency, Amplitude ^3^)	Posture
**Human studies**									
1. Turbanski et al. (2005)	RCT	N = 52 Sex M/F = 38/14 Mean age = 69.1	3.3	2 groups - WBV (26) - Moderate walking (26)	1 day	Zeptor Med System Vertical	1 session 5 × 60 s	6 Hz +/– 1 Hz/s 3 mm	N.R.
2. Haas and Turbanski et al. (2006)	Cross-over design	N = 68 Sex M/F = 53/15 Mean age = 65	2–4	2 groups - WBV → Rest (34) - Rest → WBV (34)	1 day	Zeptor Med System Vertical	1 session 5 × 60 s	6 Hz+/– 1 Hz/s 3 mm	Semi-squat
3. Haas and Buhlmann et al. (2006)	RCT	N = 28 Sex M/F = N.R. Mean age: 63.1	2–4	2 Groups - WBV (19) - Rest (9)	1 day	SRT Medical^®^ System Vertical	1 session 5 × 60 s	6 Hz N.R.	N.R.
4. Dincher et al. (2020)	RCT	N = 36 Sex M/F = 18/18 Mean age = 69.3	2.1	4 groups - 6 Hz WBV (9) - 12 Hz WBV (9) - 18 Hz WBV (9) - Placebo: (9)	1 day	Galileo Med Advanced Side-alternating	1 session 5 × 60 s	6 Hz, 12 Hz, 18 Hz 4 mm	Semi-squat
5. Dincher et al. (2021)	RCT	N = 54 Sex M/F = 24/30 Mean age = 72.5	2.11	4 groups - 6 Hz PD (10) - 12 Hz PD (10) -18 Hz PD (8) - Placebo (9)	1 day	Galileo Med Advanced Side-alternating	1 session 5 × 60 s	6 Hz, 12 Hz, 18 Hz N.R.	Semi-squat
6. Kaut et al. (2011)	RCT	N = 35 Sex M/F = 28/7 Mean age = 69.4	2.6	2 groups - WBV (18) - Sham WBV (17)	5 days	SR Zeptor Device Vertical	3 sessions 5 × 60 s	6.5 Hz N.R.	Semi-squat
7. Kaut et al. (2016)	RCT	N = 56 Sex M/F = 36/20 Mean age = 67.0	2.7	2 groups - WBV (30) - Sham vibration (26)	8 days	SR Zeptor Device Vertical	4 sessions 6 × 60 s	7 Hz 3 mm	Semi-squat
8. Ebersbach et al. (2008)	RCT	N = 21 Sex M/F = 14/7 Mean age = 73.8	N.R.	2 groups - WBV (10) - Physical therapy (11)	3 weeks	Galileo Device Side-alternating	20 sessions/ 5 sessions/wk 2 × 15 min	25 Hz 7–14 mm	Semi-squat
9. Guadarrama-Molina et al. (2020)	Non-randomized clinical trial	N = 45 Sex M/F = 27/18 Mean age = 63.5	2	3 groups - WBV (15) - Conventional therapy (15) - Combined (15)	3 weeks	Fitvibe Excel Pro Vibration Trainer Vertical	20 sessions/ 3 sessions/wk 8 × 20 s	20 Hz 2 mm	Eight active postures
10. Kapur et al. (2012)	RCT	N = 23 Sex M/F = 16/7 Mean age = 65.4	2–3	2 Groups - WBV (12) - Listening to music (11)	4 weeks	SMART Lounge, Vertical	28 sessions/ 7 sessions/wk 1 × 30 min	30–500 Hz N.R.	Sitting on a vibrating chair
11. Corbianco et al. (2018)	RCT	N = 20 Sex M/F = 20/0 Mean age = 57.9	2	2 groups: - WBV (10) - Aerobic treadmill training (10)	4 weeks	Galileo Med L2000 Side-alternating	16 sessions/ 4 sessions/wk 20 × 60 s	26 Hz 4 mm	Semi-squat
12. Arias et al. (2009)	RCT	N = 21 Sex M/F = 12/9 Mean age = 66.7	N.R.	2 groups - WBV (10) - Placebo (11)	5 weeks	N.R.	12 sessions/ 2–3 sessions/wk 5 × 60 s	6 Hz N.R.	Semi-squat
13. Gaβner et al. (2014)	RCT	N = 17 Sex M/F = 13/4 Mean age = 69.7	2.6	2 groups - WBV (8) - Placebo (9)	5 weeks	SRT Zeptor Medical Vertical	12 sessions 2–3 session/wk 5 × 60 s	6 Hz ± 1 Hz noise 3 mm	Semi-squat
**Animal studies**									
14. Zhoa et al. (2014)	RCT	N = 25 MPTP mice 2 injections of MPTP (30 mg/kg) Brains decapitated	N.R.	4 groups - MPTP LAV LF (7) - MPTP LAV HF (6) - MPTP (6) - Healthy mice, no vibration (6)	4 weeks	Columbus Instruments Vertical	28 sessions/ 7 sessions/wk 15 × 60 s	10 Hz, 20 Hz 5 mm	Not fixated

^1^ H&Y = Hoehn and Yahr; a scale used for the staging of functional disability associated with PD. A higher score indicates a greater disability. ^2^ Total number of sessions/number of sessions per week; number of bouts per session x bout duration. ^3^ Terminology of author reproduced. Abbreviations: N.R. = not reported; MPTP = neurotoxicant inducer of Parkinsonism; LAV = low-amplitude vibration; LF = low frequency; HF = high frequency.

**Table 2 biology-11-01238-t002:** Results of studies regarding the effects of WBV on PD patients or PD animal models.

Reference	Examined Domains	Outcome Measure	WBV (Mean ± SD) ^2^		Control (Mean ± SD) ^2^		WBV vs. Control	Effect Size (g) ^1^	Main Finding
			Pre	Post	Pre	Post			
**Human studies**									
1. Turbanski et al. (2005)	Postural stability	Sway reduction narrow stance (%) Sway reduction Tandem stance (%)	N.R. N.R.	↓14.9% ** ↓24% *	N.R. N.R.	–7.1% * –11.3%	ns. *p* = 0.04	–	“Random whole-body vibration can improve postural stability in PD but these effects depend on the test condition (narrow position vs. tandem stance)”.
2. Haas and Turbanski et al. (2006)	Motor symptoms	UPDRS-III (%change)	N.R.	WBV→Rest ↓16.8% ** Rest→WBV ↓14.7% **	N.R.	N.R.	*p* < 0.01 *p* < 0.01	–	“As the treatment was connected each time with significant improvements in the UPDRS motor score while the control condition led to small, insignificant changes only, one has to conclude that the treatment has beneficial effects on PD motor symptoms”.
3. Haas and Buhlmann et al. (2006)	Proprioceptive performance	Minimum knee angle (°) Maximum knee angle (°)	103.1 ± 4.9 125.6 ± 5.7	103.5 ± 3.9 126.5 ± 7.2	101.9 ± 5.4 124.1 ± 5.1	100.6 ± 4.0 126.5 ± 5.4	ns. ns.	–0.36 –0.24	“This study did not identify changes in proprioceptive performance after short-term mechanical training stimuli that reduced PD symptoms and especially postural control disturbances”.
4. Dincher et al. (2020)	Flexibility Freezing	Sit and reach test (best of 3) 360° turn test combined (s)	6 Hz: –11 ± 13.0 12 Hz: –5.4 ± 15.3 18 Hz: –5.8 ± 13.1 6 Hz: 8.8 ± 3.0 12 Hz: 8.4 ± 2.7 18 Hz: 6.3 ± 1.6	6 Hz: –9.25 ± 13.3 12 Hz: –6.5 ± 15.9 18 Hz: 1.9 ± 10.8 * 6 Hz: 8.4 ± 2.7 12 Hz: 7.3 ± 2.7 18 Hz: 5.8 ± 1.0	–12.0 ± 8.8 12.3 ± 7.3	–11.2 ± 11.8 12.6 ± 8.5	N.R. N.R. N.R. N.R. N.R. N.R.	0.08 –0.14 0.58 0.11 0.23 0.13	“It could be shown that higher frequencies seem to achieve a greater improvement from pretest to posttest than lower frequencies”.
5. Dincher et al. (2021)	Balance Stability	95% ellipse of sway (cm^2^) Anterior–posterior stability (cm) Medio-lateral stability (cm)	6 Hz: 451.6 ± 196.15 12 Hz: 555 ± 624.16 18 Hz: 333.5 ± 138.3 6 Hz: –18.7 ±16.5 12 Hz: –16.4 ± 13.6 18 Hz: –20.01 ± 12.4 6 Hz: –6.05 ± 10.8 12 Hz: –4.5 ± 9.4 18 Hz –2.3 ± 9.8	6 Hz: 499.9 ± 202.7 12 Hz: 562 ± 602.2 18 Hz: 211.5 ± 143 * 6 Hz: –23.2 ± 17.3 12 Hz: –18.9 ± 12.5 18 Hz: –23.2 ± 17.3 6 Hz: –2.1 ± 8.3 12 Hz: –6.9 ± 8.0 18 Hz: –3.6 ± 10.8	388 ± 188.2 –14.8 ± 11.2 –1.3 ± 9.8	536.5 ± 264.8 –15.4 ± 14.1 –3.4 ± 8.6	N.R. N.R. N.R. N.R. N.R. N.R. N.R. N.R. N.R.	0.45 0.29 1.33 0.25 0.14 0.18 0.61 –0.03 0.08	“WBV can cause an increase in the sway area and an improvement to anterior–posterior center displacement. Vibration frequency seems to play a subordinate role”.
6. Kaut et al. (2011)	Motor symptoms	UPDRS-III sum score	26.9 ± 10.4	19.8 ± 8.5 *	24.4 ± 9.4	25.0 ± 11.0	N.R.	0.76	“A significant number of responders was found for bradykinesia and postural stability. The extent of improvement of bradykinesia in the treatment group was evident in comparison to the sham-treated group and baseline”.
7. Kaut et al. (2016)	Postural stability Mobility Fall risk Gait Motor symptoms	Mean sway (mm) TUG (s) Tinetti score 8 MW (s) UPDRS-III sum score	356.5 ± 212.1 11.1 ± 10.1 23.1 ± 4.9 6.7 ± 2.6 24.8 ± 13.4	293.9 ± 144.5 * 8.3 ± 3.7 24.2 ± 3.3 * 6.1 ± 1.9 * 18.9 ± 12.5 **	272.0 ± 59.9 12.2 ± 10.6 21.6 ± 7.4 6.7 ± 3.3 25.4 ± 10.5	263.9 ± 64.3 10.5 ± 7.7 21.9 ± 7.6 6.5 ± 3.0 21.4 ± 10.8 **	ns. ns. ns. ns. ns.	0.39 0.13 0.13 0.15 0.16	“Stochastic resonance therapy significantly enhanced postural stability even in individuals with an increased risk of falling. Thus it offers a potential supplementation to canonical treatments of PD”.
8. Ebersbach et al. (2008)	Balance Postural stability Gait Motor symptoms	Tinetti score Mean sway (mm) Time walk 10 m (s) UPDRS-III sum score	9.3 ± 3.1 1937.0 ± 1250.0 17.6 ± 5.0 23.0 ± 4.9	12.8 ± 1.9 ** 1306.0 ± 331.0 15.1±3.5 ** 17.6 ± 4.5 **	8.3 ± 2.9 1832.0 ± 746.0 18.4 ± 4.2 25.9 ± 8.1	11.5 ± 2.4 ** 2256.0 ± 681.0 16.5 ± 2.5** 16.9 ± 5.0 **	ns. ns. ns. ns.	0.11 1.24 0.15 –0.59	“Equilibrium and gait improved in patients with PD receiving conventional WBV or PT after treatment and follow up. There was no conclusive evidence for superior efficacy of WBV compared with conventional balance training”.
9. Guadarrama-Molina et al. (2020)	Balance	Berg balance scale (score)	- WBV 47.3 ± 4.1 - CT + WBV 45.3 ± 4.5	- WBV 51.3 ± 2.6 * - CT + WBV 51.13 ± 3.4 *	- CT 48.0 ± 2.9	- CT 51.3 ± 3.4 *	ns. p = 0.02	0.21 0.68	“Rehabilitation therapy, either conventional, WBV or combined, improved functional balance in patients with PD. Combined therapy had a greater improvement compared to conventional therapy”.
10. Kapur et al. (2012)	Motor symptoms Non-motor symptoms Depression Anxiety Fatigue	MDS-UPDRS-III score MDS-UPDRS I score BD1 score ISQ anxiety FFS score	36.3 ±9.0 8.3 ± 3.9 6.7 ± 4.6 0.8 ± 1.0 29.9 ± 9.2	25.6 ± 14.7 * 6.9 ± 3.4 * 5.2 ± 3.9 * 0.5 ± 0.5 * 26.0 ± 12.0 *	41.7 ± 9.8 9.5 ± 5.9 7.4 ± 4.9 1.1 ± 1.2 28.6 ± 14.1	34.2 ± 13.5 * 6.7 ± 3.2 * 5.0 ± 2.3 * 0.7 ± 1.0 * 23.8 ± 12.3 *	ns. ns. ns. ns. ns.	0.26 –0.32 –0.21 –0.10 –0.07	“Auditory sensory stimuli with relaxation in the lounge chair has equivalent benefit as vibration on motor function and mental state”.
11. Corbianco et al. (2018)	Recovery phase Metabolic effects	RER Free fatty acids Branched-chain AAs	0.90 ± 0.07 N.R. N.R.	0.87 ± 0.05 N.R. ↑ *p* < 0.05 *	0.87 ± 0.05 N.R. N.R.	0.75 ± 0.04 * ↑ *p <* 0.01 * ↑ *p <* 0.01 *	N.R. N.R. N.R.	1.56 – –	“Both exercise groups, were significantly consumed BCAAs, whereas free Trp, the serotonin precursor, increased. The WBVT does not appear to require a long recovery time and leads to feeling less fatigued, whereas AER needs an appropriate recovery time after the training session”.
12. Arias et al. (2009)	Gait stability Motor symptoms Balance	Gait velocity (m/s) FRT (mm) UPDRS III score Berg balance scale (score)	0.7 ± 0.2 207.3 ±74.7 27.8 ± 7.5 44.1 ± 8.7	0.90 ± 0.2 ** 324.1 ± 51.7 ** 23.0 ± 6.8 * 48.4 ± 7.4 **	N.R. 221.4 ± 73.6 N.R. N.R.	N.R. 257.2 ± 72.5 * N.R. N.R.	ns. ns. ns. ns.	1.13	“There was no difference between the experimental (vibration) and placebo groups in any outcomes. These results suggest that reported benefits of vibration are due to a placebo response”.
13. Gaßner et al. (2014)	Motor symptoms Mobility Stability Gait Balance	UPDRS-III score TUG (s) FRT (m) Step–walk–turn (s) One-leg test (s)	29 ± 14 11 ± 2.5 0.89 ± 7.4 8.9 ± 1.2 18.9 ± 14.4	27 ± 13 9.0 ± 2.2 * 0.92 ± 8.4 7.45 ± 1.5 * 31.5 ± 17.1 *	19 ± 7 10.1 ± 1.6 0.91 ± 3.9 8.03 ± 1.8 29.4 ± 18	18 ± 6 9.7 ± 1.3 * 0.89 ± 3.9 7.29 ± 1.7 * 40.8 ± 19.6*	ns. p = 0.041 p = 0.004 ns. ns.	0.09 0.79 0.01 0.42 0.07	“In most of the parameters, a significant interaction of the main outcome measure “time∗group” could not be established. An intervention with random WBV could lead to effects similar to a placebo treatment”.
**Animal studies**									
14. Zhoa et al. (2014)	Dopamine in striatum BDNF levels striatum	HLPC analysis of dopamine (ng dopamine/mL) Enzyme-linked immunosorbent assay (pg/mL)	– – – –	- MPTP LAV LF 205 ± 66.4 - MPTP LAV HF 206.4 ± 22.0 - MPTP LAV LF 25.0 ± 0.3 - MPTP LAV HF 25.3 ± 0.9	– – – –	- MPTP 128.3 ± 38.5 - MPTP 24.6 ± 0.2	*p <* 0.01 *p <* 0.01 *p <* 0.05 *p <* 0.05	1.29 2.30 0.99 1.43	“Data show that four weeks of vibration training almost completely prevented the MPTP-induced loss of DA neurons in the substantia nigra and reduction in DA levels in the striatum and an upregulation of BDNF”.

^1^ Hedges’ g effect sizes with a negative value indicate a decrease in performance following WBV vs. control; effect sizes with a positive value indicate an increase in performance following WBV vs. control. ^2^ If the standard error was reported, the standard deviation was calculated with: standard deviation = standard error*√n. Abbreviations: N.R. = not reported; ns. = non-significant; TUG = timed up and go test; 8 MW = 8-m walking; UPDRS-III = unified Parkinson’s Disease rating scale motor scores; BDI = Beck depression inventory; ISQ = status questionnaire; FFS = fatigue severity scale; FRT = functional reach test; RER = respiratory exchange ratio; MPTP = neurotoxicant inducer of Parkinsonism; LAV = low-amplitude vibration; LF = low frequency; HF = high frequency; BDNF = brain-derived neurotrophic factor; CT = conventional therapy. * *p <* 0.05 significance versus pre-test data. ** *p <* 0.01 significance versus pre-test data.

**Table 3 biology-11-01238-t003:** General overview of study characteristics regarding potential ameliorating effects of WBV on neuropathological mechanisms of PD in animals and humans (N = 18 studies).

Reference	Design	Target Population	Sample	Groups (n)	Intervention Duration	Vibration Protocol			
						Type (device, vibration)	Temporal aspects ^1^	Intensity (frequency, amplitude ^2^)	Posture
**Outcome measure: neurotransmitters**									
**Human studies**									
15. Goto and Takamatsu (2005)	Cross-over design	Healthy, young men	N = 8 Sex M/F = 8/0 Mean age = 23.4	1 group - WBV + Control (8)	2 weeks	Galileo 900 Side- alternating	2 sessions/ 1 session/wk 10 × 60 s	26 Hz Q2.5 mm	Static squat position
**Animal studies**									
16. Okada et al. (1983)	RCT	Wistar rats	N = 64 Sex M/F = 64/0 Brains decapitated	2 groups - WBV (32) - Placebo (32)	5 h	EMIC 505 Vertical	1 session 1 × 240 min	20 Hz N.R	Not fixated
17. Ariizumi and Okada (1985)	RCT	Wistar rats	N = 8 Sex M/F = 8/0 Brains decapitated	2 groups - WBV (4) - Placebo (4)	5 h	Emic 505 Vertical	1 session 1 × 240 min	20 Hz N.R.	Not fixated
18. Nakamura et al. (1992)	RCT	Wistar rats	N = 10 Sex M/F = 10/0 Brains decapitated	2 groups - WBV (5) - Noise (5)	1 day	505-D: EMIC Vertical	1 session 1 × 90 min	20 Hz N.R	Not fixated
19. Heesterbeek et al. (2017)	RCT	C57BI/6J mice	N = 14 Sex M/F = 14/0	2 groups - WBV (7) - Pseudo WBV (7)	5 weeks	LEVELL R.C. Vertical	25 sessions/ 5 sessions/wk 1 × 10 min	30 Hz 40–60 µm, 29–75 µm, 14–54 µm	Not fixated
**Outcome measure: inflammatory markers and neurotrophic factors**									
**Human studies**									
20. Ribeiro et al. (2018)	RCT	Women with fibromyalgia (FM) and healthy women (HW)	N = 40 Sex M/F = 0/40 Mean age = 51.6	2 groups - FM+ WBV (20) - HW + WBV (20)	1 day	FitVibe Vertical	1 session 8 × 40 s	40 Hz 4 mm	Active squats
21. Jawed et al. (2020)	Cross-over design	Young and old adults	N = 11 Sex = N.R. Mean age young = 24 Mean age old = 55	1 group - WBV + standing - WBV + squat - Squatting	2–3 weeks	Power Plate Vertical	3 sessions/ 1 session/wk 8 × 60 s	35 Hz 4 mm	Dynamic squatting and standing
22. Rodriguez-Miguelez et al. (2015)	RCT	Elderly subjects	N = 28 Sex M/F = 8/20 Mean age = 70.7	2 groups - WBV exercise training program (14) - Daily routine (14)	8 weeks	Fitvibe Vertical	16 sessions/ 2 sessions/wk 4 × 30–45–60 s	20–35 Hz 4 mm	Static and dynamic squats
23. Simao et al. (2019)	RCT	Elderly women with knee osteoarthritis	N = 15 Sex M/F = 0/15 Mean age = 3	2 groups - WBV +squats (7) - Squats (8)	12 weeks	FitVibe Vertical	36 sessions/ 3 sessions/wk 6 × 20 s, 8 × 40 s	35–40 Hz 4 mm	Active squats
**Animal studies**									
24. Raval et al. (2018)	RCT	Senescent female rats + artery occlusion	N = 12 Sex M/F = 0/12 Blood sample	2 groups - WBV (6) - No WBV (6)	6 weeks	N.R.	30 sessions/ 5 sessions/wk 2 sessions/day 2 × 15 min	40 Hz N.R	Not fixated
25. Wu et al. (2018)	RCT	Apolipoprotein E-deficient mice (atherosclerosis)	N = 16 Sex M/F = 16/0 Blood sample	2 groups - WBV (8) - No WBV (8)	12 weeks	Huanzhen Machinery Limited Company Vertical	72 sessions/ 6 sessions/wk 1 × 10–30–60–120 min	15 Hz 2 mm	Not fixated
**Outcome measure: brain-related changes**									
**Human studies**									
26. Choi et al. (2019)	Cross-over design	Healthy male adults	N = 18 Sex M/F = 18/0 Mean age = 23.4	1 group - WBV 27 Hz - WBV 20 Hz - WBV 10 Hz - WBV 0 Hz	1 day	Galileo^®^ Advanced Plus Side-alternating	1 session 8 ×/30 s	0 Hz, 10 Hz, 20 Hz, 27 Hz 4 mm	Slight squat position
**Animal studies**									
27. Huang et al. (2018)	RCT	Sprague Dawley rats with cerebral ischemia	N = 115 Sex M/F = 115/0	3 groups - WBV (50) - No WBV (50) - No cerebral ischemia and WBV (15)	4 weeks	N.R.	20 sessions/ 5 session/wk 1 × 30 min	15 Hz 5 mm	Not fixated
28. Boerema et al. (2018)	RCT	C57Bl/6J mice	N = 20 Sex M/F = 20/0	2 groups - WBV (10) - Pseudo-WBV (10)	5 weeks	LEVELL R.C. Oscillator + Power Amplifier Vertical	27 sessions/ 5 sessions/wk 1 × 10 min	30 Hz 0.0537 mm	Not fixated
29. Peng et al. (2021)	RCT	Chronic restraint stress rat model (CRS)	N = 18 Sex M/F = 18/0 Mean age = 3 months	3 groups - CRS (6) - CRS + WBV (7) - Control (5)	8 weeks	ZB series- 0977234 Side- alternating	48 sessions/ 6 sessions/wk 1 × 30 min	30 Hz 4.5 mm	Not fixated
30. Cariati et al. (2021)	RCT	Wild-type BALB/c male mice (infectious disease)	N = 32 Sex M/F = 32/0 Brains decapitated	4 groups - Young mice WBV (12) - Old mice WBV (12) - Young mice no WBV (4) - Old mice no WBV (8)	12 weeks	Power Club Vigarano Mainard Vertical	36 sessions/ 3 sessions/wk 5 × 3 min, 3 × 2 min	45 Hz 1.5 mm	Not fixated
**Outcome measure: oxidative stress**									
**Human studies**									
31. Santos et al. (2019)	Experimental matched case–control study	Women with fibromyalgia (FM) and healthy women (HW)	N = 42 Sex M/F = 0/42 Mean age = 51.1	2 groups - FM (21) - HW (21)	1 day	FitVibe, Gyma- Uniphy Vertical	1 session 8 × 40 s	40 Hz 4 mm	Squatting exercises
**Animal studies**									
32. Liu et al. (2016)	RCT	Db/db mice (diabetes type 2)	N = 24 Sex M/F = 0/24 Mean age = 8 weeks	3 groups - Db/db WBV (8) - Db/db (8) - No Db/db and WBV(8)	12 weeks	N.R.	84 sessions 1 × 60 min	45 Hz N.R.	Not fixated

^1^ Total number of sessions/number of sessions per week; number of bouts per session × bout duration. ^2^ Terminology of author reproduced. Abbreviation: N.R. = not reported.

**Table 4 biology-11-01238-t004:** Results of studies regarding potential ameliorating effects of WBV on neuropathological mechanisms of PD.

Reference	Examined Domains	Outcome Measure	WBV (Mean ± SD) ^2^	Control (Mean ± SD) ^2^	WBV vs. Control	Effect Size (g) ^1^	Main Findings
**Outcome measure: Neurotransmitters**							
**Human studies**							
15. Goto and Takamatsu (2005)	Epinephrine Noradrenaline	Plasma epinephrine (pg/mL) Plasma norepinephrine (pg/mL)	pre 26.7 ± 15.3 post 38.0 ± 14.1 * pre 288.0 ± 109.5 post 456.4 ± 254.3 *	– –	– –	0.73 0.81	“A single bout of a WBV session enhanced acute epinephrine and norepinephrine secretion in the blood”.
**Animal studies**							
16. Okada et al. (1983)	Norepinephrine	Norepinephrine in hypothalamus (ng/g) Norepinephrine in hippocampus (ng/g)	post 896.7 ± 461.0 post 664.8 ± 578.7	post 1891.3 ± 1291.5 post 1176.1 ± 1349.7	*p <* 0.01 ns.	–1.01 –0.49	“WBV caused a decrease in cerebral noradrenalin. The decrease does not occur in the brain generally, but only in particular regions. The hypothalamic content of norepinephrine was most affected, but there was a tendency for norepinephrine content to decrease in the hippocampus”.
17. Ariizumi and Okada (1985)	Cerebral neurotransmitters	Norepinephrine in hypothalamus (ng/g) Dopamine cortex (n/ng) Dopamine striatum (n/ng) Serotonin cerebellum (ng/g) Serotonin in hypothalamus (ng/g)	post 885.4 ± 154.8 post 823.0 ± 102.4 post 332.6 ± 349.8 post 578.3 ± 156.6 * post 2879.7 ± 756.4 *	post 1882.5 ± 447 ** post 631.1 ± 89.6 post 742.0 ± 198 post 289.2 ± 144.6 post 1633.0 ± 361	*p <* 0.01 ns. ns. *p <* 0.05 *p <* 0.05	–2.59 1.73 –1.25 1.67 1.83	“Norepinephrine in the whole brain and especially in the hypothalamus is a better indicator of vibration exposure than serotonin, and norepinephrine is affected by the intensity but not the frequency of vibrations. Noradrenalin and serotonin in the hypothalamus change in the opposite direction. Dopamine concentrations in the brain are basically unaffected by vibration”.
18. Nakamura et al. (1992)	Cerebral dopamine systems in several regions of the brain	Dopamine protein (nucleus accumbens) (ng/mg) Homovanillic acid/dopamine ratio (frontal cortex) (ng/mg)	post 44.9 ± 36.4 post 0.337 ± 0.06	post 9.4 ± 2.4 post 0.204 ± 0.06	*p* = 0.016 *p* = 0.032	1.24 2.00	“These results suggest that the responses of organisms via acute whole-body vibrations may be critically mediated by cerebral dopamine systems, in particular, by the mesocortical dopamine system.”
19. Heesterbeek et al. (2017)	ChAT- immunoreactivity	Chat-corrected optical density in SS cortex Chat-corrected optical density in basolateral amygdala	post 0.23 ± 0.02 post 0.33 ± 0.03	post 0.19 ± 0.02 post 0.27 ± 0.03	*p <* 0.05 *p <* 0.01	1.87 1.87	“The results of this study reveal that the positive effects of WBV on attention may be (at least in part) mediated by an increased activity of the NBM cholinergic system. WBV could therefore be a suitable intervention strategy in conditions where a reduced cholinergic forebrain activity plays a role”.
**Outcome measure: Inflammatory markers and neurotrophic growth factors**							
**Human studies**							
20. Ribeiro et al. (2018)	Inflammatory markers Growth factors	Adiponectin (pg/mL) sTNFR1(pg/mL) sTNFR2 (pg/mL) Plasma BDNF (pg/mL)	- HW + WBV: pre 35,977.6 ± 2239.5 post 39,660.1 ± 4926.8 - FM + WBV: pre 43,342.7 ± 1343.7 post 38,102 ± 895.6 * - HW + WBV: pre 697.3 ± 115.1 post 998.7 ± 281.4 * - FM + WBV: pre 1014.8 ± 153.4 post 845.0 ± 63.9 * - HW + WBV: pre 2179.7 ± 296.6 post 2000.9 ± 28.5 - FM + WBV: pre 2179.7 ± 247 post 1789.1 ± 222.3 * - HW + WBV: pre 1696.4 ± 446.5 post 1778.3 ± 446.8 - FM + WBV: pre 1689 ± 329.2 post 1563.0 ± 305.8	– – – – – – – –	– – – – – – – –	0.94 –4.50 1.37 –1.42 –0.83 –1.63 0.18 –0.39	“A single acute session of mild and short WBV can improve the inflammatory status in patients with fibromyalgia (FM), reaching values close to those of matched healthy women (HW) at basal status. The neuroendocrine mechanism seems to be an exercise-induced modulation towards greater adaptation to stress response in these patients”.
21. Jawed et al. (2020)	Inflammatory markers Growth factors	Interleuking 6 (pg/mL) TNF-a (pg/mL) Interleuklin 10 (pg/mL) Vascular endothelial growth factor (VEGF) (pg/mL)	- Standing + WBV ^3^: pre 24.8 ± 12.6 post 19.4 ± 9.3 pre 21.2 ± 12.9 post 29.8 ± 16.3 * pre 43.1 ± 12.60 post 57.8 ± 16.9 * pre 252.1 ± 12.6 post 269.5 ± 15.3 *	- Squat: pre 19.8 ± 11.6 post 23.4 ± 10.3 pre 29.8 ± 20.6 post 29.9 ± 21.2 pre 44.6 ± 18.6 post 46.5 ± 19.2 pre 253.7 ± 9.6 post 258.2 ± 15.3	N.R. N.R. N.R. N.R.	–0.78 0.45 0.72 0.93	“WBV has the potential to positively influence inflammation. Significant increases in TNF-α,VEGF, and IL-10 only occurred during vibration alone, although IL-6 approached a significant drop with vibration alone, with no differences detected with age”.
22. Rodriguez-Miguelez et al. (2015)	Inflammatory markers	TLR2 (% content) TLR4 (%content) TNFa (arbitrary units)	pre 94.3 ± 34.2 post 59.9 ± 40 * pre: 107.8 ± 43.8 post 60.1 ± 22.5 * pre 2.9 ± 1.1 post 1.7 ± 0.8 *	pre 99.7 ± 34.4 post 94.3 ± 40.0 pre 100.1 ± 15.3 post 97.1 ± 35.5 pre 2.9 ± 1.1 post 3.0 ± 0.75	*p <* 0.05 *p <* 0.05 *p <* 0.05	–0.76 –1.39 –1.33	“WBV counteracts, at least in part, age-related low-grade chronic inflammation. This response seems to be mediated by a downregulation of the TLR2 and TLR4 MyD88- and TRIF-dependent signaling pathways”.
23. Simao et al. (2019)	Growth factors	Plasma BDNF (%Δ)	pre 4.78 post +4.2%	pre 3.0 post –32.5%	*p <* 0.05	–	“The addition of WBV to squat-exercise training improves lower-limb muscle performance in elderly women with knee osteoarthritis, likely by increasing BDNF, suggestive of a modulation in neuromuscular plasticity”.
**Animal studies**							
24. Raval et al. (2018)	Inflammatory markers Growth factors	Caspase 1 (% region) Interleukin 10 (% region) ASC (% region) ^b^ BDNF (% region) pTrK-B (% region)	post 77.4 ± 8.2 post 67.1 ± 24.8 post 84.9 ± 7.4 post 165.7 ± 18.9 post 133.0 ± 10.0	post 165.6 ± 12.6 post 215.0 ± 11.0 post 141.1 ± 6.1 post 107.0 ± 8.0 post 72.6 ± 13.8	*p <* 0.05 *p <* 0.05 *p <* 0.05 *p <* 0.05 *p <* 0.05	–7.66 –7.12 –7.65 3.73 4.63	“WBV induces a significant reduction in inflammatory markers and infarct volume with significant increases in brain-derived neurotrophic factor and improvement in functional activity after tMCAO in middle-aged female rats that were treated with WBV as compared to the non-WBV group”.
25. Wu et al. (2018)	Inflammatory markers Growth factors	Relative protein level IL-6/GAPDH IGF-1 (ng/mL)	post 0.4 ± 0.3 post 167.7 ± 91.3	post 0.8 ± 0.3 post 272.0 ± 95.3	*p <* 0.05 *p <* 0.05	–1.26 –1.06	“The levels of IGF-1 in serum and expression of IL-6 protein in mice aorta decreased significantly in the WBV group compared to control”.
**Outcome measure: brain-related changes**							
**Human studies**							
26. Choi et al. (2019)	Cortical activation during different frequencies of WBV	FNIRS results of: OxyHb concentration 10 Hz OxyHb concentration 20 Hz OxyHb concentration 27 Hz	N.R.	N.R.	*p <* 0.05	–	“The results from the present study show that oxyHb concentrations of the motor, prefrontal, and somatosensory cortex areas are higher during the 27 Hz vibration condition than the control or 10 Hz conditions”.
**Animal studies**							
27. Huang et al. (2018)	Neurogenesis	Neu/BrdU-labelled cells in cortex	- Ischemia + WBV: 14 d: post 4.0 ± 13.4 21 d: post 13.9± 21.9 28 d: post 28.0± 26.9	- Ischemia: 14 d: post 3.0 ± 7.4 21 d: post 9.9 ± 21.2 28 d: post 18.9 ± 35.4	ns. *p <* 0.001 *p <* 0.001	0.09 0.18 0.29	“WBV promoted neurogenesis after long-term exposure after cerebral ischemia in rats.”
28. Boerema et al. (2018)	Brain glucose uptake	F-FDG uptake (%ID/g)	pre 3.8 ± 0.7 post 3.9 ± 0.7	pre 3.7 ± 0.7 post 4.0 ± 0.7	ns. ns.	–0.27	“The 18F-FDG PET data does not reveal any significant difference in brain uptake ratio due to WBV. There was a small but not significant increase in the pseudo WBV group post-treatment”.
29. Peng et al. (2021)	Neurons Neural degeneration Neurotropic factors	Neun (n of surviving neurons) F-Jade C (% of control) IGF-1 (ng/mL) BDNF (ng/mL)	- CRS + WBV: post 86.0 ± 16.4 - CRS + WBV: post 125.9 ± 63.8 - CRS+ WBV: post 43.9 ± 11.1 - CRS + WBV: post 676.1 ± 46.6	- CRS: post 62.9 ± 14.7 - Control ^4^: post 109.2 ± 18.8 - CRS: post 870.5 ± 704.5 - Control^4^: post 80.4 ± 53.9 - CRS post 27.4 ± 14.5 - Control ^4^ post 41.7 ± 9.4 - CRS: post 506.3 ± 107.5 - Control ^4^ post 831.2 ± 327.1	*p <* 0.05 *p <* 0.05 *p <* 0.05 *p <* 0.05	1.37 –1.45 1.20 1.97	“WBV could reverse behavioral dysfunction, inhibit the degeneration of neurons, alleviate the damage of neurons and the pathological changes of glial cells, enhance trophic factor expression, and ameliorate the downregulation of dendritic and synaptic proteins after CRS. The effect of WBV in rats may be mediated via the reduction in hippocampal neuronal degeneration and by improving expression of synaptic proteins”.
30. Cariati et al. (2021)	Hippocampal synaptic plasticity	%PS amplitude	- Young mice + WBV: pre 101.4 ± 3.1 post 386.8 ± 87.3 * - Old mice + WBV: pre 101.4 ± 3.1 post 466.9 ± 151.4 **	- Young mice: pre 100.3 ± 2.0 post 325.2 ± 53.6 * - Old mice: pre 101.5 ± 2.0 post 249.5 ± 90.4 *	ns. *p <* 0.05	0.99 2.25	“Vibratory training can modulate synaptic plasticity differently, depending on the protocol used, and that the best effects are related to the training protocol characterized by a low vibration frequency and a longer recovery time (3 × 150 s, 45 Hz WBV)”.
**Outcome measure: Oxidative Stress**							
**Human studies**							
31. Santos et al. (2019)	Oxidative stress markers	TBARS FRAP SOD CAT	- HW + WBV: post 0.14 ± 0.13 - FM + WBV: post 0.2 ± 0.18 * - HW + WBV: post 180.0 ± 103.6 - FM + WBV: post 239.1 ± 82.9 * - HW + WBV: post 1.9 ± 0.09 * - FM + WBV: post 1.1 ± 0.18 * - HW + WBV: post 31.2 ± 14.2 * - FM + WBV: post 2.6 ± 1.3	- HW: post 0.2 ± 0.18 - FM: post 0.8 ± 1.4 - HW: post 188.3 ± 78.8 - FM: post 485.6 ± 208.5 - HW: post 1.1 ± 1.8 - FM: post 0.9 ± 0.91 - HW: post 7.5 ± 17.8 - FM: post 3.8 ± 8.7	ns. *p <* 0.05 ns. *p <* 0.05 *p <* 0.05 *p <* 0.05 *p <* 0.05 *p <* 0.05	–0.37 –0.59 –0.09 –1.52 0.62 0.30 1.44 –0.19	“A single trial of WBV exercise improved all oxidant and antioxidant parameters towards a greater adaptation to the stress response in women with fibromyalgia (FM) as compared to the healthy women group (HW)”.
**Animal studies**							
32. Liu et al. (2016)	Oxidative stress	GSH (µmol/L) GSH-Px (u/mgprotein)	- Db/db + WBV: post 282.5 ± 69.3 - Db/db + WBV: post 923.8 ± 9.6	- Db/db: post 195.5 ± 24.0 - Db/db: post 880.5 ± 156.7	*p <* 0.05 ns.	1.59 0.37	“WBV attenuates oxidative stress to ameliorate liver steatosis and thus improves insulin resistance in db/db mice. Therefore, WBV administration is a promising treatment for individuals who suffered from central obesity and IR”.

^1^ Hedges’ g effect sizes with a negative and positive values indicate a (relative) decrease and increase, respectively, in the mean score of the parameter concerning WBV (vs. control). ^2^ If the standard error was reported, the standard deviations were calculated with: standard deviation = standard error*√n. ^3^ In the WBV + squatting group, no significant differences in pre–post-tests were observed for all parameters. ^4^ Significant differences were observed between the control and CRS groups for all parameters. Abbreviations: BDNF = brain-derived neurotrophic factor; FM = fibromyalgia; sTNFR1 = soluble tumur necrosis factor receptor 1; sTNFR2 = soluble tumor necrosis factor receptor 2; TLR2 = Toll-like receptor 2; TLR4 = Toll-like receptor 4 TNFa = tumor necrose factor 1; pTrK-B = tropomyosin-related kinase B receptor; IL-6/GAPDH = interleukin 6/glyceraldehyde 3-phosphate dehydrogenase; IGF-1 = insulin-like growth factor-1; OxyHb = oxygen hemoglobin; F-FDG = fludeoxyglucose; Neu/BrdU = neuron/bromodeoxyuridine; F-Jade C = fluoro-jade stain C; TBARS = thiobarbituric acid reactive substances; FRAP = ferric reducing ability of plasma; SOD = superoxide dismutase antioxidant enzyme activity; CAT = catalase; GSH = glutathione; CRS = chronic restraint stress model; ChAT = choline acetyltransferase. * *p* < 0.05 significance versus pre-test data. ** *p* < 0.01 significance versus pre-test data.

**Table 5 biology-11-01238-t005:** Recommendations for practice and further research of WBV in relation to PD.

Recommendations for Practice	Recommendations for Further Research
Apply WBV if moderate-to-high conventional exercise not possible or additional to conventional exercise	High-quality research with sufficient duration (≥3 weeks), session frequency (≥3 sessions/week) and vibration frequency (≥20 Hz)
At least three sessions per week	RCTs with contrasting control groups
Vibration frequency of at least 20 Hz	RCTs with different levels for frequency and/or peak-to-peak displacement
Start under adequate supervision	Trials with side-alternating WBV
	Add cognitive function and other non-motor variables affected by PD to outcomes
	Animal research using PD mouse models and/or PD-disease-relevant cellular models
	Improve reporting on WBV studies using guidelines (van Heuvelen et al., 2021)

## Data Availability

Not applicable.

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
