# Peer review of "Potential of Whole-Body Vibration in Parkinson’s Disease: A Systematic Review and Meta-Analysis of Human and Animal Studies"

_biology, 2022, doi:10.3390/biology11081238_

Round 1
Reviewer 1 Report
Dear authors, thank you very much for this interesting work. Please see the following comments to improve the article:
In the methods section, please note as an inclusion criterion the publication dates (e.g. from January 1990 to April 2022).
Further in the methods section, you should shortly describe the assessment instrument in all studies found. What are their psychometric properties? This could be an interesting fact for more discussion points.
Please see/ add the following studies (why did you not find them?):
Dincher, A., & Wydra, G. (2021). Chapter 3: Effect of Whole Body Vibration on Balance in Parkinson's Disease: A Randomized Controlled Pilot Study. In MedDocs Publishers (eds.), Alzheimer's Disease and Treatment (Vol. 3, pp. 21-25).
Line 378: reference #50 double mentioned
Reference list: references #46, #56, and #60 are the same
The font size in the whole document is not consistent and should be corrected.
Best regards!
Author Response
Please, see the attachment

Reviewer 2 Report
The manuscript by Arauz and colleagues provides a systematic review on the potential benefits of whole body vibration (WBV) on Parkinson's disease patients. Authors reviewed a diverse and large compilation of papers, from which they concluded that WBV still displays inconclusive results and should be better investigated in the future. I have no major issues with the manuscript, which is nicely written, however I will only make some minor suggestions:
Lines 75-76. The authors propose to assess WBV as an alternative to exercise. However, I think the particular choice of WBV should be better justified.
Line 77. I think the manuscript will benefit considerably by including a paragraph describing the nature of the WBV technique, especially for the benefit of readers who are unaware of WBV or of its particular health benefits;
Line 101. I suggest "systematic" instead of "comprehensive";
Line 111. I suggest to remove "comprehensive";
Line 122. Please change the format of the citation;
Line 145. The authors should uniformize the way they refer to themselves - either as "authors" or as "reviewers";
Table 1. What is the meaning of "H&Y"?;
Tables 2-4. Why is some text highlighted in Bold?
Table 3. In the study of "Wu et al. (2018)" it should be "atherosclerosis" and not "artherosclerosis";
Table 3. In the study of "Gotoo & Taka-matsu" I suggest to replace "noradrenalin/noradrenaline" with "norepinephrine";
Author Response
Please, see the attachment

Reviewer 3 Report
This manuscript presents a systematic review of theWhole-Body Vibration in patients with Parkinson Disease:. The manuscript is generally well written, but it is unclear what this review adds to what is already known and have been published earlier. No clear research question seems to be formulated, the conclusions are unclear and other major concerns with this manuscript that make is necessary major changes for publication.
My specific comments are stated below. Overall, several important issues need to be addressed and some are of methodological character which requires a considerable revision of the paper.
1. I recommend formulating a specific research question and performing a meta-analysis to answer the question.
2. The title of this manuscript is very long. Perhaps a more concise version for clarity, interes and ease of read.
3. KEYWORDS: Please use recognised MeSH terms as this will assist others when they are searching for information on your research topic. The following website will provide these (simply start typing in a keyword and see if it exists or find an alternative if it does not): https://www.ncbi.nlm.nih.gov/mesh
4. Introduction: I suggest that background should be improved, with more details about foot implications and the vibration in patients parkinson disease, see the research of Jimenez et al https://pubmed.ncbi.nlm.nih.gov/35581151/ and Navarro et al https://pubmed.ncbi.nlm.nih.gov/34275724/
5. Materials and Methods: This section is very well written, meets the requirements for systematic reviews (the Authors employed a comprehensive search strategy, identified all relevant studies, assessed the results for inclusion/exclusion and for quality and finally presented summary of findings) and fill the gaps in scientific knowledge. Now, you need to include the ID number of PROSPERO related with your was recorded in the International Prospective Register of Systematic Reviews on health sciences associate topics. The inclusion and exclusion criteria appear sound.
Thus, the authors have not performed a systematic review, according to international standards PRISMA Guidelines, so they do not provide specific numerical data.
6. Results: Please, provide clear results was using the tool Review Manager (RevMan) of The Cochrane Library, v.5.3 and describe them. Use appropriate statistics
7. Within your discussion, outline your results, discuss their novelty and their application to practice.
8. Conclusions need to be softened, modified a in order to reflect only the study findings.
Author Response
Please, see the attachment

Round 2
Reviewer 3 Report
I think the authors addressed the concerns. It is ready for publication.